# Integrated, Automated, Fast PCR System for Point-Of-Care Molecular Diagnosis of Bacterial Infection

**DOI:** 10.3390/s21020377

**Published:** 2021-01-07

**Authors:** Dongkyu Lee, Deawook Kim, Jounghyuk Han, Jongsu Yun, Kang-Ho Lee, Gyu Man Kim, Ohwon Kwon, Jaejong Lee

**Affiliations:** 1Daegu Research Center for Medical Devices, Korea Institute of Machinery and Materials, Daegu 42994, Korea; dongkyu@kimm.re.kr (D.L.); wooki@kimm.re.kr (D.K.); hyek9210@kimm.re.kr (J.H.); jsyoon@kimm.re.kr (J.Y.); kangholee6@kimm.re.kr (K.-H.L.); 2Department of Mechanical Engineering, Kyungpook National University, Daegu 41566, Korea; gyuman.kim@knu.ac.kr; 3Nano-Mechanical Systems, Korea Institute of Machinery and Materials (KIMM), Daejeon 34103, Korea

**Keywords:** fast PCR, automated system, molecular diagnostics, sample preparation, rapid thermocycle

## Abstract

We developed an integrated PCR system that performs automated sample preparation and fast polymerase chain reaction (PCR) for application in point-of care (POC) testing. This system is assembled from inexpensive 3D-printing parts, off-the-shelf electronics and motors. Molecular detection requires a series of procedures including sample preparation, amplification, and fluorescence intensity analysis. The system can perform automated DNA sample preparation (extraction, separation and purification) in ≤5 min. The variance of the automated sample preparation was clearly lower than that achieved using manual DNA extraction. Fast thermal ramp cycles were generated by a customized thermocycler designed to automatically transport samples between heating and cooling blocks. Despite the large sample volume (50 μL), rapid two-step PCR amplification completed 40 cycles in ≤13.8 min. Variations in fluorescence intensity were measured by analyzing fluorescence images. As proof of concept of this system, we demonstrated the rapid DNA detection of pathogenic bacteria. We also compared the sensitivity of this system with that of a commercial device during the automated extraction and fast PCR of *Salmonella* bacteria.

## 1. Introduction

The polymerase chain reaction (PCR) can detect molecules with higher accuracy than immunoassays, and therefore has become a standard technology for the diagnosis of viral and pathogenic infections [1,2,3]. On-site field testing of emergency molecular diagnosis is critical to quickly identify infected patients and thereby slow the spread of infectious diseases. However, the transfer of target samples to a clinical laboratory and the execution of analytical steps usually require a long time. In general, point-of-care (POC) molecular detection is impractical because skilled operators and appropriate equipment are not readily available in the field. Therefore, practical rapid molecular diagnostics in the field require automated sample preparation and reduced analysis time [4,5,6,7].

The DNA from targets is usually extracted in several steps, including lysis, capture, separation, and purification. Commercialized automation systems have been implemented clinically, but are expensive, complex, and too large for application in POC testing [8,9,10]. Therefore, DNA extraction for a POC test is still performed manually. However, the process is labor-intensive due to the large number of pipetting steps, and is therefore slow and subject to human error [11,12]. To solve these problems, many attempts have been made to assemble low-cost, portable, automated systems from 3D-printed parts, off-the-shelf electronics, and smartphones [13,14,15,16,17]. However, POC testing requires an integrated system for practical use.

DNA amplification takes the most time in the PCR process. In general, 40 cycles take 1–2 h due to the low energy-transfer rate at a typical reaction volume of 25–50 μL during thermal ramp cycles from 60 °C to 95 °C. Commercial PCR systems have improved the DNA amplification speed using optimized thermal ramp cycles on copper sample plates and thermoelectric devices. However, reducing the DNA amplification time to <1 h is not sufficient for POC testing. Several studies have achieved 30 cycles in 3–15 min by using various thermalization methods such as serpentine microfluidic heat exchanging, optical (LED (light emitting diode) or laser) heating, and thin-film resistor heating [18,19,20,21,22]. 

Reducing the sample volume is a simple way to further increase the rate of DNA amplification by accelerating heat-energy transfer. However, controlling micro-flow in the microfluidic system or loading small drops into a small reaction chamber require technical adaptation by operators who are skilled in the use of a large volume system that uses conical microtubes. In addition, a small sample volume reduces sensitivity, increases sample-concentration variations, and entails complex system integration [23]. 

To speed up the thermal cycle for large sample volumes, preheated water switching and mechanical shuttling or rotating of heat sources have been used to change only the sample temperature without changing the temperature of the entire heating system [22,23,24,25,26,27,28]. However, the preheated water bath system uses a large bath and complex pump and valves, and it is difficult to integrate with automation and detection steps, so it is not applicable to PCR devices in POC tests. 

In this study, a low-cost, compact, integrated, rapid molecular detection system was developed by using inexpensive 3D-printed parts, commercial electronics and motors for the application of POC testing. The system performs automated sample preparation, fast PCR, and fluorescence intensity analysis within 20 min, despite the large sample volume (50 μL). Fast thermal ramp cycles were obtained using a customized thermocycler that automatically transports the sample between heating and cooling blocks. The variations in fluorescent intensity were monitored by analyzing the brightness of fluorescent images. We used the system to achieve fast PCR detection of *Salmonella* bacteria. 

## 2. Experimental Section

### 2.1. Materials

*Salmonella* was cultured in Luria-Bertani(LB) broth (Sigma-Aldrich, St. Louis, MO, USA) at 37 °C for 24 h. Pre-mixed solutions for general PCR were purchased from Bioneer (Cat No. S-1010, Seoul, Korea) and pre-mixed solutions for fast PCR were purchased from Applied Biosystems (Cat No. 4444556, Temecula, CA, USA). Lysis buffer (Cat No. 67563), elution buffer (Cat No. 19077) and magnetic beads (400 nm, Cat No. 1026883) were obtained from a thermoscientific preparation kit (Qiagen, Valencia, CA, USA). Phosphate-buffered saline (PBS) solution was purchased from Corning (NY, USA). Proteinase K (Cat No. BP1700-100) was purchased from Thermo Fisher (Carlsbad, CA, USA), and washing buffers I (Cat No. CMB-007) and II (CMB-005) were purchased from Cosmogentech (Seoul, Korea). DNA was extracted and separated by following the manufacturer’s protocols. Primer DNA of *Salmonella* was purchased from Bioneer (Seoul, Korea); the sequences were: (forward) 5′ AGC GTA CTG GAA AGG GAA AG 3′ (20mer), (reverse) 5′ ATA CCG CCA ATA AAG TTC ACA AAG 3′ (24mer), (probe) 5′ [FAM] CGT CAC CTT TGA TAA ACT TCA TCG CA [BHQ1] 3′ (26mer). For the automated fast PCR system, many parts of the commercial electronics and motors were purchased from several electronics shopping sites. In addition, the detailed electronic parts and motors are described in the Appendix A.

### 2.2. DNA Manual Extraction Protocol

Microbes (10^8^ CFU/mL *Salmonella*) were spiked after being separated by centrifuging. Firstly, 100 µL of the target sample in 5 mL PBS buffer was mixed with 200 µL of lysis buffer, then 20 µL of proteinase K (20 mg/mL) was added and reacted for 5 min at room temperature with manual shaking to obtain high-molecular-weight DNA. To capture DNA, the solution was mixed with 20 µL of a suspension of magnetic beads (20 mg/mL) and 200 µL of isopropanol, then stirred for 3 min using a pipette mixing. After the magnetic beads were separated by a magnet, the DNA captured by the magnetic beads was washed twice with 200 µL of washing buffer I and II to remove proteins and impurities from the solution. Then, 200 µL of elution buffer was added to collect the extracted target DNA. Manual extraction takes around 20 min. The same reagent solutions of DNA extraction were used in both manual and automated sample preparations (Appendix A). 

### 2.3. DNA Purity and Amplification Detection

The concentration and purity of the extracted DNA in both manual and automated sample preparations were measured using a spectrophotometer (Bibby Scientific, Staffordshire, UK). The DNA extraction purity was determined from the ratio of absorbance *A* at 260 and 280 nm, compared to absorbance at 320 nm as (A_260_–A_320_)/(A_280_–A_320_). Nucleotides, RNA, and DNA absorb at 260 nm, whereas protein absorbs at 280 nm. To compare the extraction efficiencies of DNA extracted using either manual or automated methods, the amplification of the extracted DNA was quantified using a commercial PCR device (Applied Biosystems, Temecula, CA, USA). Firstly, 50 µL of PCR reagent was prepared in general conical PCR microtubes for large-volume PCR amplification. Then, 50 µL of PCR reagent was mixed with 25 µL of PCR master mix, 2 µL of forward primer (10 pM/µL), 2 µL of reverse primer (10 pM/µL), 2 µL of probe dye (10 pM/µL), and 19 µL of target DNA. An annealing/elongation temperature of 60 ± 2 °C and a denature temperature of 93 ± 2 °C were chosen, and the annealing and elongation times were optimized by reference to the variations in fluorescence intensity. The cycle threshold (Ct) in the variations in fluorescence intensity for both of the methods was measured while thermal cycles in the two steps of standard PCR processes were repeating. Rapid thermal ramp cycles from 60 °C to 93 °C were generated by transporting conical PCR microtubes between pre-heated (105 °C) and pre-cooled blocks (10 °C). Temperature changes were measured using a glass-coated thermistor (Th310J39GBSN, Ampenol advanced sensor) in the microtube without DNA for the development of a temperature prediction algorithm. 

## 3. Results and Discussion

### 3.1. Automated Fast PCR System

The automated fast PCR system (Figure 1) was fabricated from 3D-printed parts, commercial electronic elements and motors. The system (Figure 1a) has X-Z axis translational stages. Two servo motors were implemented to separately move a magnetic bar and magnet cover in the Z-axis direction. A magnetic bar and a temperature sensor were fixed at the ends of the tips in servo motor 1. A magnet plastic cover was mounted on servo motor 2. The maximum rotation angle of the servo motor is 160°, and the rotational angle is translated to linear motion. The moveable distance in the Z-axis is ~45.3 mm, and the minimum length of movement in the Z-axis was calculated to be ~400 μm per degree. A step motor with a linear guide can move the Z-axis motor stage in the X-axis direction. The distance between two pitches of the pulleys was ~187 mm, and the length of timing belts was calculated to be ~413 mm, using timing belt calculation formulas. The minimum length of movement in the X-axis was calculated to be ~235 μm per step, using the diameter of the pulley and the step angle of 1.8°. 

We designed the integrated sample cartridge to consecutively perform all PCR procedures (automated DNA extraction, fast DNA amplification, and visual inspection; Figure 1b). To achieve fast DNA amplification, both heating and cooling blocks were located in the cartridge holder. The heating block was maintained at 105 °C using a ceramic heater, and the cooling block was maintained at 10 °C using a thermoelectric cooler. The fluorescence image can be monitored using a blue LED, 466 nm/520 nm filters, and a complementary metal oxide semiconductor (CMOS) camera. The fluorescence intensity variations can simply be obtained from the fluorescence image before and after completing 40 cycles for POC testing. The assembled automated fast PCR system (Figure 1c) measures 25 cm × 18 cm × 23 cm (length × width × thickness), it weighs <1 kg, and it can be powered by a 12 V, 3.5 A power adapter, and is therefore portable and suitable for use in POC testing. The designed system accommodates two tubes for the proof-of-concept study, but an additional extra tube array can be expanded to complete control tests of molecular analysis. 

### 3.2. Automated Processes of Sample Preparation 

The magnetic particles (MPs) were used to extract *Salmonella* DNA because they can easily separate and purify the extracted DNA. The automated extraction processes basically follow the manual protocol. We prepared a cartridge that has four wells, containing lysis buffer, washing I, washing II, and elution buffer. The magnetic tip was mounted at the end of servo motor 1, and the magnet cover was mounted at the end of servo motor 2 (Figure 2a). The workflows of mixing, capturing, transport, and dispersion were automated (Figure 2b). Firstly, the target sample was mixed with lysis reagent solution in the first well. The magnet cover can be moved up and down by servo motor 2 and be quickly shaken from side to side by the step motor for mixing and MP dispersion. Then, MPs with attached DNA were collected by the magnetic tip, transported to the washing chamber, washed, then re-dispersed in the solution by mixing without the magnetic tip. Finally, the extracted DNA was released in the elution buffer (Figure 2c). To increase the purity and concentration of the extracted DNA, MPs should be fully dispersed in the solution. We found that the rapid shaking (10 Hz) of the magnet cover from side to side using the step motor was the key motion. 

MP transport and re-dispersion to the next well entails several steps (Figure 2c). A magnetic cover stage (Z2) moves down, then a magnetic tip stage (Z1) moves down. When MPs with DNA were collected by the magnetic tip, only Z2 moved up. When the Z-axis stage transports to the next well, Z1 moves up after only Z2 moves down. Then, MPs can be re-dispersed into the washing solution of the next well. The total time required for the automated DNA extraction is ~5 min due to the rapid shaking mix without turnaround time. However, manual extraction takes ~20 min due to longer mixing and waiting times between each step, and this can be dramatically increased if a large number of samples must be prepared [16].

*Salmonella* DNA was extracted by both the automated and manual methods. The initial concentration and purity of the extracted DNA from the automated and manual methods were measured using a spectrophotometer (Figure 3a); at 10^8^ CFU/mL *Salmonella*, the concentration was ~25 ng/µL with purity of 1.68 by the manual method, and ~20 ng/µL with purity of 1.66 for DNA by the automated method. The manual method achieved a higher average DNA concentration and purity than the automated system did, but measurements by the manual method also had three times higher standard deviation in purity and ten times higher standard deviation in concentration than the automated method.

To confirm the efficiencies of DNA extraction by the manual and automated methods, PCR amplification detection of the samples was compared using the commercial PCR device. The real-time curves (Figure 3b) increased faster for the manually treated samples than for the automated methods; the manually treated samples had the cycle threshold (Ct) ~ 17 and the automatically treated samples had Ct ~ 18.3. However, Ct deviations of the manual extraction sample were four times larger than those of the automated extraction sample; this result means that the automated extraction is more reproducible and reliable than manual extraction.

### 3.3. Fast PCR 

Pre-heated (105 °C) and pre-cooled (10 °C) blocks were implemented (Figure 4) for fast PCR amplification. Rapid thermal ramp cycles from 60 °C to 93 °C were generated by transporting conical PCR microtubes between hot and cold blocks (Figure 4a). Microtubes containing DNA (positive) and without DNA (negative) were mounted on Z2 and moved between the hot block (105 °C) and the cooling block (10 °C). Temperature changes were measured using a thermistor (Th310J39GBSN, Ampenol advanced sensor) in the microtube. Thermal cycles were rapid in the large sample (50 µL) (Figure 4b); 40 cycles could be completed in 8 min (black line), 10.5 min (red line), 13.8 min (blue line), and 17 min (green line) when the annealing/elongation time between 58 and 62 °C was 1, 5, 10, and 15 s. The maximum heating rate was 10.2 °C/s, and the maximum cooling rate was 16.5 °C/s (Figure 4c). The annealing/elongation temperature was maintained by radiant heat obtained by controlling the distance between the microtube and the heating block before the microtube was fully inserted into the heating block. A single ramp cycle takes 13, 17, 22, and 27 s at an annealing/elongation time of 1, 5, 10, and 15 s, respectively. A commercial PCR device takes ~ 38 min to amplify DNA for 40 cycles with 1 s of denaturation and 10 s of annealing/elongation at the same conical PCR microtube and the sample volume (50 µL) because the maximum heating and cooling rates are 2.3 °C/s. Thus, the rates of temperature change were > 5 times higher in our fast PCR system than in the commercial device. The effects of sample volume on the thermocycler speed were investigated (Appendix A). For a sample of 20 µL volume, 40 cycles could be completed in 12.8 min. The small reduction in amplification time is attributed to limited heat conduction though the conical microtube. If the heating block temperature is >105 °C, variations in the fluorescence intensity decrease due to the denaturation of Taq DNA polymerase (Appendix A). It means that important variables such as the heating and cooling temperatures, reaction chamber shape, and sample volume should be considered for the fast thermocycle. In addition, the efficiency of gene amplification is affected by the annealing/elongation time. To optimize PCR amplification, variations in the fluorescence intensity of both fast PCR and general PCR reagents at different annealing/elongation times were measured using the commercial PCR system before and after completing 40 cycles in the developed fast PCR system. The optimal annealing/elongation time was 10 s for fast PCR and 20 s for general PCR (Figure 4d).

The fluorescence images were monitored by a CMOS camera. Blue light (passed through a 466 nm bandpass filter) was shone on the sample, and the CMOS camera captured 520 nm filtered fluorescence images. The fluorescence image was monitored and analyzed using a single-board microcomputer (Raspberry Pi). The captured image was cropped to a size of 280 × 300 pixels, then the total brightness value was calculated by summing the brightness values of all pixels (Appendix A). The amplification process was stopped repeatedly to obtain fluorescence images every five cycles. The fluorescence image changed from dark green to light green during PCR amplification every five cycles (Figure 5a). The brightness of positive and negative samples varied over cycles (Figure 5b); the fluorescence intensity of the positive sample strongly increased compared to that of the negative sample.

Although the automated fast PCR system could not determine Ct values precisely because the images were captured only every fifth cycle, Ct was estimated to be 15 to 20 considering the intensity variations of the negative sample. The system is suitable for fast screening during POC testing.

The relative fluorescence intensity (Δ*I* − *I*(0))/(*I*(max) − *I*(0) increased as amplification time increased (Figure 5c). The PCR duration for 40 cycles was reduced to 13.8 min with the fast PCR system from 38 min with the commercial PCR device. The PCR duration can be further reduced to <10 min for 40 cycles if the reaction chamber shape can be changed or the amplification volume of samples can be reduced. The performance comparison between commercial and developed systems is described in the Appendix A.

To confirm the utility of the automated fast PCR system, real-time PCR curves were analyzed using a commercial PCR device on 10-fold serial dilutions of the automatically-extracted DNA sample from this system (10^−1^ to 10^−5^, 8 to 0.0008 ng/µL) (Figure 6a). As dilution increased, the Ct increased and the maximum change in fluorescence intensity at 40 cycles decreased. The logarithm plot of DNA concentration versus Ct (Appendix A) is linear. The linear trend is generally utilized for quantification analysis. In the case of the fast PCR system, however, qualitative analysis can be performed by the variations in fluorescence intensity at 40 cycles. To compare the detection sensitivity between the automated fast PCR system and the commercial PCR device, fluorescence intensity at 40 cycles from both systems showed log-linear trends. The detection sensitivity of the automated fast PCR system and the commercial PCR device were calculated to be 10^−3^ and 10^−4^ ng/µL, respectively. The detection sensitivity of DNA amplification is affected by the fluorescence optical setup. The sensitivity of the proposed device may be improved by optimizing the fluorescence optical setup by using a high-resolution camera.

Real-time PCR curves were obtained for *Salmonella*, *E. coli O157* and negative samples. The PCR assay for detection of *Salmonella*
*species* was not reactive to *E. coli O157* and the negative sample. Variations in fluorescence intensity from the same *Salmonella* PCR assay were obtained using the commercial PCR device (Figure 7a) and the automated fast PCR system (Figure 7b). In both cases, the *Salmonella* DNA showed large variations in fluorescence intensity and a light green image, whereas the *E. coli O157* DNA and the negative sample showed small variations in fluorescence intensity and dark green images. Thus, the automated fast PCR system can be used for rapid screening of target DNA in POC testing.

## 4. Conclusions

An affordable, compact, integrated, and automated fast PCR system was developed to perform automated sample preparation and fast PCR in POC testing. The automated fast PCR system was assembled from inexpensive 3D-printed parts, commercial electronics and motors. DNA detection using PCR involves a series of procedures, including sample preparation, amplification, and fluorescence intensity analysis. The automated system completed DNA sample preparation, including the extraction, separation and purification processes, in ≤5 min. The automated extraction was much more efficient than manual extraction. The variation in measurements was clearly smaller after automated sample preparation than after manual DNA extraction. We also designed a customized thermocycler to transport a sample between heating and cooling blocks. The rapid two-step PCR amplification performed 40 cycles within ~13.8 min, despite the large sample volume (50 μL). Variations in fluorescence intensity were measured by analyzing the fluorescence images obtained by an optical setup that used a CMOS camera and a blue LED. As proof of concept, we demonstrated the rapid DNA detection of *Salmonella*. Despite the short testing period ≤20 min, the DNA detection efficiency of this system was comparable to that of the commercial device. This system can be utilized for automated fast molecular detection without experts, for emergency POC testing of infectious viruses and pathogenic bacteria.

## Figures and Tables

**Figure 1 sensors-21-00377-f001:**
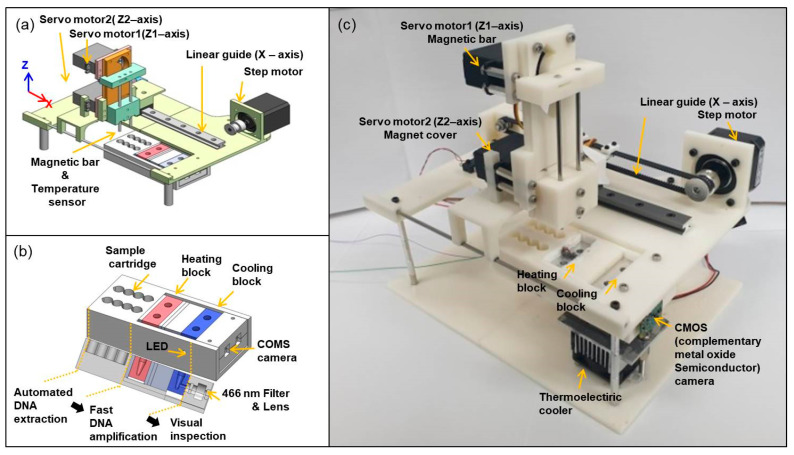
Schematic images of (**a**) whole system and (**b**) sample cartridge holder with cross-sectional view. (**c**) Photograph of automated fast PCR system.

**Figure 2 sensors-21-00377-f002:**
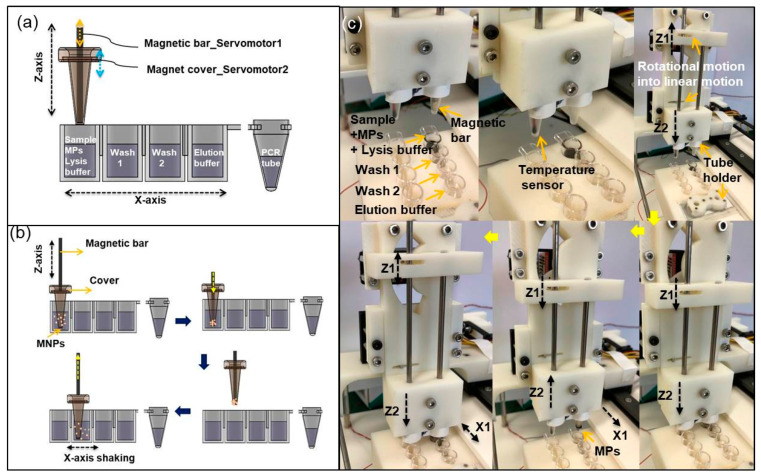
Schematic images of (**a**) cross-sectional view of a sample cartridge and (**b**) automated workflows including mixing, capturing, transport, and dispersion. The cartridge has four wells: lysis buffer, washing I, washing II, and elution buffer for automated sample preparation, and a conical PCR microtube for DNA amplification. (**c**) Photographs of each step for magnetic particle (MP) transport and re-dispersion to next well.

**Figure 3 sensors-21-00377-f003:**
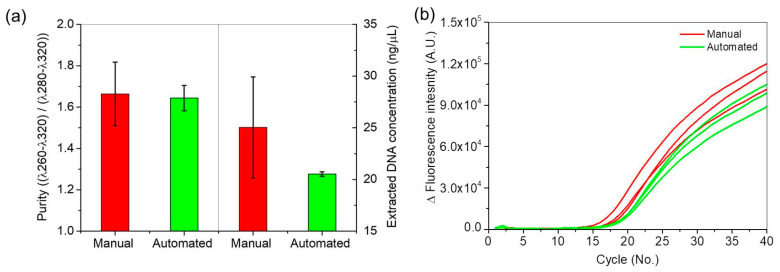
(**a**) DNA purity and concentration of manual and automated extraction samples, and (**b**) variations in fluorescence intensity of manual and automated extraction samples during PCR amplification.

**Figure 4 sensors-21-00377-f004:**
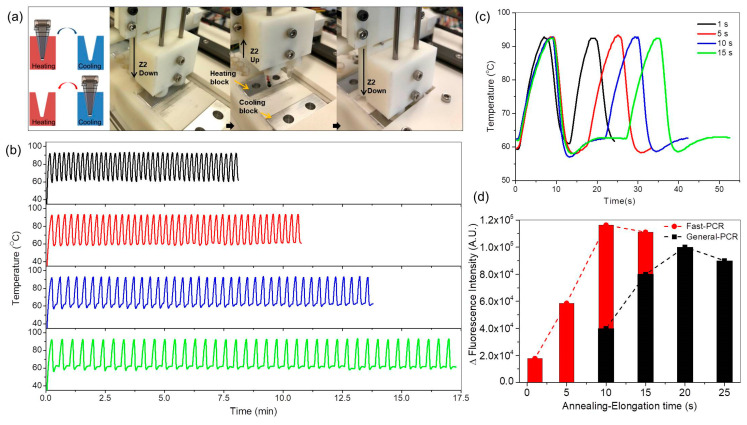
(**a**) Schematic image and photographs of each step to generate rapid thermal ramp cycles by heating exchangers transporting conical PCR microtubes between heating and cooling blocks. (**b**) Temperatures during thermocycling for fast PCR amplification when the annealing/elongation time was 1 s (black line), 5 s (red line), 10 s (blue line), and 15 s (green line), (**c**) temperatures of two thermal cycles depending on annealing/elongation time for enlargement. (**d**) Optimization of annealing/elongation time for fast PCR (red bar) and general PCR (black bar).

**Figure 5 sensors-21-00377-f005:**
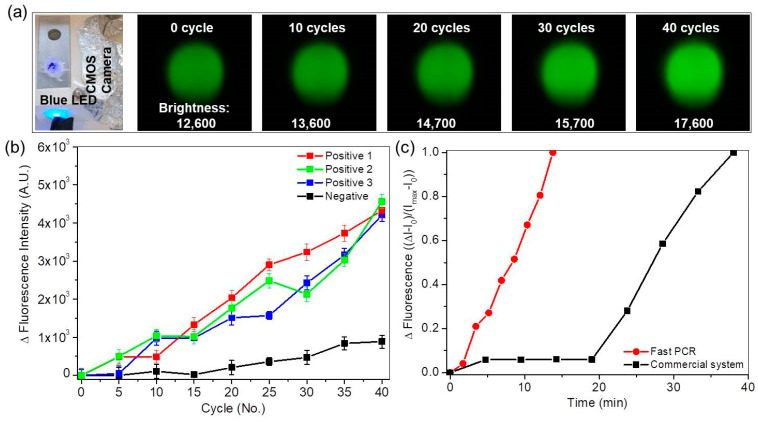
(**a**) Photographs of fluorescence optical setup and captured fluorescence images during PCR amplification every fifth cycle. Fluorescence images change from dark green to light green during PCR amplification. Numbers on fluorescence images are calculated brightnesses. (**b**) Variations in brightness of positive and negative samples with respect to cycles. (**c**) Relative fluorescence intensity ((Δ*I* − *I*(0))/(*I*(max) − *I*(0)) with respect to amplification time for the fast PCR system (red) and the commercial PCR device (black).

**Figure 6 sensors-21-00377-f006:**
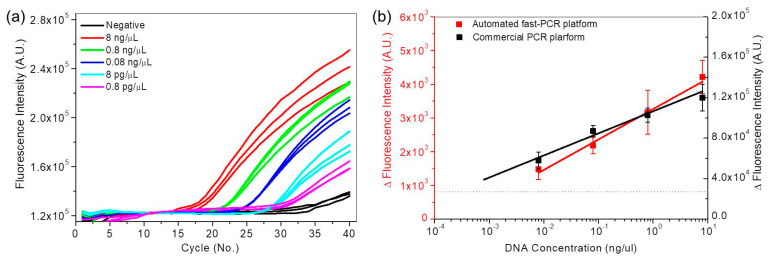
(**a**) Real-time fluorescence curves of 10-fold serial dilution samples (10^−1^ to 10^−5^, 8 to 0.0008 ng/ µL) from the commercial PCR device. (**b**) Variations in fluorescence intensity at 40 cycles from the automated fast PCR system and commercial PCR device.

**Figure 7 sensors-21-00377-f007:**
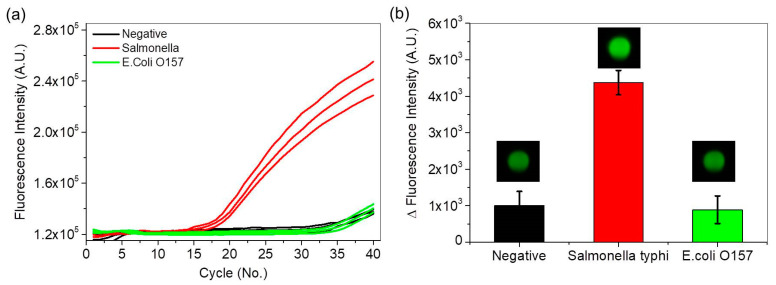
(**a**) Real-time PCR curves for *Salmonella* (red lines), *E. coli O157* (green lines), and negative samples (black lines) using the commercial PCR device, and (**b**) variations in fluorescence intensity and images during the same *Salmonella* PCR assay using the automated fast PCR system.

## Data Availability

Data available on reasonable request from the corresponding author.

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
