# Peer review of "Integrated, Automated, Fast PCR System for Point-Of-Care Molecular Diagnosis of Bacterial Infection"

_sensors, 2021, doi:10.3390/s21020377_

Round 1

Reviewer 1 Report

The paper presents a very interesting concept for an inexpensive and fast automated sample preparation and PCR amplification based on 3d printed parts and commercially available electronics. The system is shown to work under experimental and proof of concept conditions, but a number of problems should be addressed before publication:

  1. Materials and Methods Section needs improvement:
    1. The system is ill described. The purpose of Materials and Methods section is to allow other to reproduce experiments and results. However, the system is only described in too general terms, without any details about the actual 3D printer models, or even electronics specifications. These should at least be part of the supplementary Information. As is, the paper results cannot be reproduced by others, leaving the paper unavailable for independent scrutiny.
    2. Commercial kits and equipment must be described in further detail, indicating the kits name, reference, manufacturer and country of fabrication. For example no details were given for the pre-mixed solutions obtained from Applied Biosystems (Page 2, lines 76-77). The same is true about the solutions referred in lines 77-78 of page 2. Furthermore, these solutions are said to be obtained from Thermoscientific, but are said to be fabricated by Qiagen (Line 78), but these are two different companies. These omissions and confusions must be clarified. Also, the model of the commercial equipment used must be stated (line 103) as characteristics of commercial PCR systems vary widely, and companies like Applied Biosystems have released, over the years, a very large and diverse array of commercial equipment.
    3. No details were given on the use of the thermal sensor. Description in line 196 gives the impression that a thermal sensor (thermistor) is inserted in the PCR tube, which could potentially lead to sample cross contamination between runs and force a copper connector to be included puncturing the tube lid. This, and the efficiency of the PCR tube lid closure (to prevent leaks during denaturing at high temperature) should be discussed as it could render the system useless for diagnostic purposes. If the thermistor was only used during testing and development to derive algorithm based temperature prediction, this should be clearly stated and described.
  2. Apparatus design issues:
    1. The authors develop a system with the capacity to perform real-time data capture (shown at a 5-cycle interval), but at the end resort to end-point result evaluation, without any explanation for the downgrading. One would expect that upgrading to one cycle interval would be goal, but this was never even addressed by authors. The PCR community has long known that end-point PCR can be very misleading. PCR, due to its inherent exponential amplification curve can have dramatic changes over the course of very few cycles. Thus, resorting to end-point result gathering has always been a solution only when quantitative data was irrelevant and when real-data was not available. It is thus not easy to understand the authors choice to ignoring the real-data gathered and instead to go to end-point results. This must at least be discussed.
    2. The authors design only accommodates a 2-tube run. This is insufficient, as every PCR run should accommodate a negative and a positive control. In some settings, positive controls can be substituted by internal controls, but this requires at least double wavelength and double fluorescent probes in the system. The described system does not accommodate for either and thus a 3-tube setting is the minimum required for the testing of any sample (sample plus negative and positive controls). Although this is a proof-of-concept study, the addition of an extra tube can put stress on the electronics processing capacity (50% increased need for processing power) that may invalidate the whole design. This should at least be discussed.
  3. Results:
    1. Figure 5b describes PCR kinetics in the developed system, but results are collected only at 5 cycles intervals. Explanation on why that was the case should be given, as the PCR kinetics is known to change dramatically in only 2-3 cycles at the exponential transition stage
    2. Figure 5b shows an almost linear amplification, dramatically different from the expected exponential amplification (see figure 6a for example). This seems to indicate a very inefficient reaction maybe associated with the presence of the magnetic particles in the PCR tube. This fact has profound implications for the capacity of the apparatus to detect very diluted samples as are frequently clinically relevant samples. Despite its critical implications on the usefulness of the equipment, efficiency of the reaction was not assayed, and this linear amplification was not discussed. This is a critical issue for the acceptance of the paper.
    3. Figure 5c is inappropriate. The algorithm used to change the Fluorescence (effectively the percentage of maximum tube fluorescent) implies that all tubes will show a growing curve, even negative samples as shown. This does not reflect any biological or physical relevant property of the reaction and should thus be removed. Also, fluorescence intensity relative to reaction time is not a relevant or meaningful way of showing PCR kinetics as it will prevent kinetic comparisons between PCR reactions with different cycle timings.
    4. Line 232 cites Fig.S2, but should be citing Fig S3
    5. Figure 6 is strongly misleading and should be avoided altogether. PCR kinetics and Ct versus concentration linearity are shown for the commercial assay, but not for the developed equipment (Fig 6a and 6b). This behaviour of qPCR is known for decades, and needs not demonstration for commercial equipment. However, demonstration of this behaviour, over a large range of concentrations for the developed assay is a must needed result that was not presented or discussed using the proven Ct/Cp methods. Instead authors chose to rely on the known faulty end-point strategy. This may work for some samples and not others, due to the exponential, saturable nature of PCR. In this regard, one might argue that, under a limited set of concentrations (that may vary from target to target) it could work due to the linear response observed in figure 5b. Also, it could be argued that the strategy may work for purely qualitative results (positive/negative). If that is the aim, the attempt to prove quantitative value as in figure 6 should be removed from the paper. Furthermore, the strategy of the authors to superimpose two different graphs, using two different arbitrary Fluorescence units is flawed and should not be done. Arbitrary units can only be compared among themselves, attempt at anything else if flawed and irrelevant.
  4. Minor spelling and sentence corrections should be done:
    1. On line 86 Salmonella are said to be injected. This is the wrong verb and should be changed to something like “added”, “spiked” for example.
    2. At the end of line 91 “by” is missing. Sentence should be “the DNA-captured by the magnetic beads”.
    3. Phrase at line 178 should be reformulated.
    4. Legend to figure 4: “generate” instead of “generated”
    5. Line 194: “Microtubes containing” (remove of)
    6. Line 209 should be rephrased.
    7. Annealing/elongation should not have a space after the slash (lines 215 and 217)

Author Response

[Authors’ Replies]

ID: sensors-1017123: "Integrated, automated, fast PCR system for point-of-care molecular diagnosis of bacterial infection" by D. Lee et. al

Journal: Sensors

We would like to thank the referee for his/her valuable comments and suggestions. Accordingly, we have made an attempt to make the presentation more clearly. We believe that as a result of referee’s suggestions, our manuscript has improved with specific changes described below. 

-Reviewer 1

The paper presents a very interesting concept for an inexpensive and fast automated sample preparation and PCR amplification based on 3d printed parts and commercially available electronics. The system is shown to work under experimental and proof of concept conditions, but a number of problems should be addressed before publication:

Materials and Methods Section needs improvement:

  1. The system is ill described. The purpose of Materials and Methods section is to allow other to reproduce experiments and results. However, the system is only described in too general terms, without any details about the actual 3D printer models, or even electronics specifications. These should at least be part of the supplementary Information. As is, the paper results cannot be reproduced by others, leaving the paper unavailable for independent scrutiny.

-We revised and added the details of electronics and motors in manuscript and supplementary information.

- In the manuscript (Page 2):

[For the automated fast PCR system, many parts of the commercial electronics and motors were purchased from several shopping sites of electronics. In addition, the detail electronic parts and motors are described in the supplementary information.]

-In the supplementary information (Page 2):

[Step motor (FL42STH33-0956A, Devicemart, Korea), Servo motor (HS-311, Hitec Rcd korea Inc., Korea), Step motor driver (A3967, twin chip), and liner bearings and slides (IGUS, Germany) were purchased to fabricate X-Z axis translational stages. Ceramic heater(CB1, Scipia, Korea) and Peltier cooler (TEC1-12706, SMG, Korea) were purchased to generate heating-cooling cycles of amplification chamber. Temperature variations were monitored by thermistor (Th310J39GBSN, Ampenol Advanced Sensor, USA). Super bright blue 3mm led (ada-301, devicemart, korea), 466nm fluorescence filter (#86-341, Edmund optics, USA), 520nm fluorescence filter (#67-016, Edmund optics, USA), Raspberry Pi3 and camera were purchased for monitoring fluorescence images. The parts of the system were designed and printed by 3D printer (Stratasys uPrinter professional desktop 3D printer).]

2.Commercial kits and equipment must be described in further detail, indicating the kits name, reference, manufacturer and country of fabrication. For example no details were given for the pre-mixed solutions obtained from Applied Biosystems (Page 2, lines 76-77). The same is true about the solutions referred in lines 77-78 of page 2. Furthermore, these solutions are said to be obtained from Thermoscientific, but are said to be fabricated by Qiagen (Line 78), but these are two different companies. These omissions and confusions must be clarified. Also, the model of the commercial equipment used must be stated (line 103) as characteristics of commercial PCR systems vary widely, and companies like Applied Biosystems have released, over the years, a very large and diverse array of commercial equipment.

-We added catalog number for the commercial kits of sample preparation and PCR solution in material section (Page 2)

  1. No details were given on the use of the thermal sensor. Description in line 196 gives the impression that a thermal sensor (thermistor) is inserted in the PCR tube, which could potentially lead to sample cross contamination between runs and force a copper connector to be included puncturing the tube lid. This, and the efficiency of the PCR tube lid closure (to prevent leaks during denaturing at high temperature) should be discussed as it could render the system useless for diagnostic purposes. If the thermistor was only used during testing and development to derive algorithm based temperature prediction, this should be clearly stated and described.

- We used the glass-coated thermistor (Th310J39GBSN, Ampenol advanced sensor). The temperature sensor inserted in the negative microtube without DNA. In addition, the thermistor insertion in the tube including punctuating the tube lid has used for the development of temperature prediction algorithm as you mentioned.

- We included the explanation in the 2.3 section (page 3, 117)

[Rapid thermal ramp cycles from 60 ℃ to 93 ℃ were generated by transporting conical PCR microtubes between pre-heated (105 ℃) and pre-cooled blocks (10 ℃). Temperature changes were measured using a glass-coated thermistor (Th310J39GBSN, Ampenol advanced sensor) in the microtube without DNA for the development of temperature prediction algorithm.]

-Apparatus design issues:

  1. The authors develop a system with the capacity to perform real-time data capture (shown at a 5-cycle interval), but at the end resort to end-point result evaluation, without any explanation for the downgrading. One would expect that upgrading to one cycle interval would be goal, but this was never even addressed by authors. The PCR community has long known that end-point PCR can be very misleading. PCR, due to its inherent exponential amplification curve can have dramatic changes over the course of very few cycles. Thus, resorting to end-point result gathering has always been a solution only when quantitative data was irrelevant and when real-data was not available. It is thus not easy to understand the authors choice to ignoring the real-data gathered and instead to go to end-point results. This must at least be discussed.

Our goal is also real-time monitoring of the fluorescence image or upgrading to one cycle interval, however, the fluorescence images is not able to be captured every one cycle. The amplification process was stopped repeatedly to obtain fluorescence images every 5 cycles.

We described these in the manuscript.

In addition, we agree that the end-point results of fluorescence intensity variation can not give quantitative analysis as reviewer was mentioned. However, our goal is the development of point-of-care testing device in field. So, it can offer qualitative analysis for the molecular diagnosis. Also, we are trying to upgrade the fluorescence imaging system.

- 4page 138lines

Although the fluorescence image is not able to be captured every one cycle, the fluorescence intensity variations can be simply obtained from the fluorescence image before and after completing 40 cycles for POCT.

- 8page 245lines

The amplification process was stopped repeatedly to obtain fluorescence images every 5 cycles.

  1. The authors design only accommodates a 2-tube run. This is insufficient, as every PCR run should accommodate a negative and a positive control. In some settings, positive controls can be substituted by internal controls, but this requires at least double wavelength and double fluorescent probes in the system. The described system does not accommodate for either and thus a 3-tube setting is the minimum required for the testing of any sample (sample plus negative and positive controls). Although this is a proof-of-concept study, the addition of an extra tube can put stress on the electronics processing capacity (50% increased need for processing power) that may invalidate the whole design. This should at least be discussed.

- We agree the review’s comments. However, this is the prototype system of fast amplification for the proof-of-concept. We described that additional extra tube array can be expanded to complete control tests of molecular analysis in 4 page 143lines.

[The designed system accommodates two tubes for the proof-of-concept study, but additional extra tube array can be expanded to complete control tests of molecular analysis.]

- Actually we are upgrading the system as blow. Let us publish the next version of prototype.

-Results:

  1. Figure 5b describes PCR kinetics in the developed system, but results are collected only at 5 cycles intervals. Explanation on why that was the case should be given, as the PCR kinetics is known to change dramatically in only 2-3 cycles at the exponential transition stage

- The fluorescence image every one cycle is not able to be captured by the optical system. The fluorescence intensity variations can be simply obtained from the fluorescence image before and after completing 40 cycles for POCT.

In Figure 5, the amplification process was stopped repeatedly to obtain fluorescence images every 5 cycles.

We described these in the manuscript in page 8 in line 245

  1. Figure 5b shows an almost linear amplification, dramatically different from the expected exponential amplification (see figure 6a for example). This seems to indicate a very inefficient reaction maybe associated with the presence of the magnetic particles in the PCR tube. This fact has profound implications for the capacity of the apparatus to detect very diluted samples as are frequently clinically relevant samples. Despite its critical implications on the usefulness of the equipment, efficiency of the reaction was not assayed, and this linear amplification was not discussed. This is a critical issue for the acceptance of the paper.

- The first reason is that we have displayed the mean intensity of three tests.

 In the individual test, the slope from 0 to15 cycles was smaller than that from 15 to 40 cycles. So, we revised the figure 5(b) with three tests

- The second reason is that the amplification process was stopped repeatedly to obtain fluorescence images every 5 cycles.

- Third reason is that sensitivity of the fluorescence imaging system was lower than commercial PCR system.

8.Figure 5c is inappropriate. The algorithm used to change the Fluorescence (effectively the percentage of maximum tube fluorescent) implies that all tubes will show a growing curve, even negative samples as shown. This does not reflect any biological or physical relevant property of the reaction and should thus be removed. Also, fluorescence intensity relative to reaction time is not a relevant or meaningful way of showing PCR kinetics as it will prevent kinetic comparisons between PCR reactions with different cycle timings.

- Although it does not reflect the biological or physical relevant properties of the reaction, In figure 5 (c), we can compare the time it takes to achieve 40 cycles in both systems using the algorithm used for fluorescence change.

- In addition, a similar algorithm was applied to reference 27. 

Ref 27: Houssin, T.; Cramer, J.; Grojsman, R.; Bellahsene, L.; Colas, G.; Moulet, H.; Minnella, W.; Pannetier, C; Leberre, M.; Plecis, A.; Chen, Y. Ultrafast, sensitive and large-volume on-chip real-time PCR for the molecular diagnosis of bacterial and viral infections. Lab Chip 2016, 16, 1401-1411.

  1. Line 232 cites Fig.S2, but should be citing Fig S3

- We revised Fig.S2 to Fig.S3.

  1. Figure 6 is strongly misleading and should be avoided altogether. PCR kinetics and Ct versus concentration linearity are shown for the commercial assay, but not for the developed equipment (Fig 6a and 6b). This behaviour of qPCR is known for decades, and needs not demonstration for commercial equipment. However, demonstration of this behaviour, over a large range of concentrations for the developed assay is a must needed result that was not presented or discussed using the proven Ct/Cp methods. Instead authors chose to rely on the known faulty end-point strategy. This may work for some samples and not others, due to the exponential, saturable nature of PCR. In this regard, one might argue that, under a limited set of concentrations (that may vary from target to target) it could work due to the linear response observed in figure 5b. Also, it could be argued that the strategy may work for purely qualitative results (positive/negative). If that is the aim, the attempt to prove quantitative value as in figure 6 should be removed from the paper. Furthermore, the strategy of the authors to superimpose two different graphs, using two different arbitrary Fluorescence units is flawed and should not be done. Arbitrary units can only be compared among themselves, attempt at anything else if flawed and irrelevant.

- We agree the reviewer’ comments. To avoid misleading of between quantitative and qualitative analysis, we moved the well-known qPCR analysis of the Figure 6(b) to supplementary information of Figure S4.

- In addition, we clarified the log-linear trends of Figure 6(c) is qualitative analysis and the purpose to compare detection sensitivity between the automated fast PCR system and the commercial PCR device in page 9 line 264.

[In case of the fast PCR system, however, qualitative analysis can be performed by the variations in fluorescence intensity at 40 cycles. To compare detection sensitivity between the automated fast PCR system and the commercial PCR device, fluorescence intensity at 40 cycles from both systems showed log-linear trends.]

Fig 6.

-Minor spelling and sentence corrections should be done:

  1. On line 86 Salmonella are said to be injected. This is the wrong verb and should be changed to something like “added”, “spiked” for example.

- We revised ‘injected’ to ‘spiked’

  1. At the end of line 91 “by” is missing. Sentence should be “the DNA-captured by the magnetic beads”.

- We added ‘by’ in the sentence.

  1. Phrase at line 178 should be reformulated.

- We revised the sentence

[To confirm the efficiencies of DNA extraction by the manual and automated methods, PCR amplification detection of the samples was compared using the commercial PCR device.]

  1. Legend to figure 4: “generate” instead of “generated”

- We revised typo error

  1. Line 194: “Microtubes containing” (remove of)

- We revised typo error

  1. Line 209 should be rephrased.

- We revised the sentence

[For a sample of 20 µL volume, 40 cycles could be completed in 12.8 min.]

  1. Annealing/elongation should not have a space after the slash (lines 215 and 217)

- We revised typo error

Reviewer 2 Report

Manuscript describes an attempt to develop the point of care PCR system.

  1. Line 18 page1 abstract: Molecule to Molecular
  2. Please rectify the text flow describing the figure. It should clear and concise.
  3. Section 2.2. Manual extraction protocol: the description is too mouthful. Missing the link from previous sentence. Authors must include a figure for this sample preparation.
  4. Please introduce a table comparing the thermos-cycler parameters of high end commercial systems in the market and your system.   
  5. Dimension of this system are comparable to the commercial PCR machines. This does not make it portable or suitable for the point of care use.
  6. Could you create figure 2d. showing sample schematics at the each step of automated sample preparation.
  7. In fig 6a; fluorescent intensity of the negative sample also goes up after 35 cycles. Could you explain this?
  8. In fig 7b, why intensity for negative control is equivalent to intensity of e-coli 0157.
  9. To qualify as a point of care testing method; you must demonstrate recognition of unknown pathogen/diagnosis of infection using this device.
  10. Or title should be modified after removing the words "point of care diagnosis".  

Author Response

[Authors’ Replies]

ID: sensors-1017123: "Integrated, automated, fast PCR system for point-of-care molecular diagnosis of bacterial infection" by D. Lee et. al

Journal: Sensors

We would like to thank the referee for his/her valuable comments and suggestions. Accordingly, we have made an attempt to make the presentation more clearly. We believe that as a result of referee’s suggestions, our manuscript has improved with specific changes described below. 

-Reviewer 2

Manuscript describes an attempt to develop the point of care PCR system.

  1. Line 18 page1 abstract: Molecule to Molecular

- We revised typo error

  1. Please rectify the text flow describing the figure. It should clear and concise.

-We revised text flow describing the moving control distance in X-Z axis stage Fig.1 in page 3~4.

  1. Section 2.2. Manual extraction protocol: the description is too mouthful. Missing the link from previous sentence. Authors must include a figure for this sample preparation.

-We add detail protocol and in the supplementary Table S1.

Table S1

Manual method

Automated method

Cell preparation

centrifugation of Salmonella cell(1*10^8 CFU) (1mL) at 9,000RPM

Removing top liquid on the cell pellet

Resuspension using 1x PBS solution(Corning Cat No. 21-040-CV)

Add Proteinase K (20μl) (Thermo fisher, Cat No. BP1700-100)

Lysis

Add Lysis buffer (200μL)(Qiagen, Cat No. 67563), mixing and reacting for 5min

Add lysis buffer (200μl) and magnet bead(20 μl) and Isopropanol(300 μl) in a tube, and then, take 200 μl and transfer it to the lysis well and moving tool for shanking mix

Bead binding

Add magnet bead(Qiagen, Cat No. 1026883) 20 μl and 300 μl of isopropanol (Sigma-Aldrich, Cat No. I9516) and pipetting mix for 3min reaction.

Separating magnetic bead

Moving magnet tool to separating magnetic bead

Remove top liquid in the tube

Moving magnet tool to Wash â…  buffer well

Washâ… 

Add 800μl of Wash â…  buffer (Cosmogenetech, Cat No. CMB-007) and vortex mixing

Shanking mix (Wash â…  buffer 200 μl)

Separating magnetic bead

Moving magnet tool to separating magnetic bead

Remove top liquid in the tube

Moving magnet tool to Wash â…¡ buffer well

Washâ…¡

Add 800μl of Wash â…¡ buffer (Cosmogenetech, Cat No. CMB-005) and vortex mixing

Shanking mix (Wash â…¡ buffer 200μl)

Separating magnetic bead

Moving magnet tool to separating magnetic bead

Remove top liquid in the tube

Moving magnet tool to Elution buffer well

Elution

Add 500μl of Elution buffer(Qiagen, Cat No. 19077) 500ul

Shanking mix (Elution buffer 200μl)

Separating magnetic bead

Remove magnetic bead

Complete DNA extraction

Complete DNA extraction

  1. Please introduce a table comparing the thermos-cycler parameters of high end commercial systems in the market and your system.   

-We add comparison between commercial system and developed system in Table S2.

- We add the sentence in page 9.

[The performance comparison between commercial and developed systems is described in the supplementary information (Table S2).]

Table S2. Comparison between commercial and developed systems.

Performance/ platform

Commercial PCR system

(Applied Biosystems)

Commercial Sample preparation machine

(Applied Biosystems)

Automated fast PCR system

(in this paper)

Sample preparation time

-

~40 min / 1-13 samples

~5 min / 2 samples

Sample preparation method

-

membrane with centrifugation

magnet bead method

Cycles / completion time

40cycles / 38min

-

40cycles / 13.5min

Denature temperature

95±1℃

-

93±2℃

Annealing/elongation temperature

60±1℃

-

62±2℃

Extension time

10 sec

-

10 sec

Heating temperature

105℃

-

105℃

Cooling temperature

-

-

10℃

Heating temperature

2.5℃

-

10.2℃

Cooling temperature

2.5℃

-

16.5℃

System size

34cm(W) × 49 cm (H) × 41cm (D)

50cm(W) × 55 cm (H) × 57cm (D)

25cm(W) × 23 cm (H) × 18cm (D)

  1. Dimension of this system are comparable to the commercial PCR machines. This does not make it portable or suitable for the point of care use.

- As shown in Table S2, the size of individual systems for PCR and sample preparation systems is larger than the developed system. In addition, shipping one system enables both sample preparation and PCR process, so the system can be adapted for POC testing.

  1. Could you create figure 2d. showing sample schematics at the each step of automated sample preparation.

- We created Table S1 instead of Figure 2d schematics because there are many steps.

  1. In fig 6a; fluorescent intensity of the negative sample also goes up after 35 cycles. Could you explain this?

- The non-specific amplification of NTC (No Template Control) often shows over 35cycles due to primer dimer as you can see reference paper below. The Ct of NTC is much higher than the Ct of template sample. The Ct of NTC can effect on detection sensitivity. However, we can delay the Ct (threshold cycle) above 35 cycle using further primer optimization.

- M Nurjayadi et al ‘Evaluation of Primers Detection Capabilities of the pef Salmonella

typhimurium Gene and the fimC Eschericia coli Gene Using Real-Time PCR to Develop Rapid Detection of Food Poisoning Bacteria’ 2018 J. Phys.: Conf. Ser. 1097 012049

  1. In fig 7b, why intensity for negative control is equivalent to intensity of e-coli 0157.
  • Figure 7b shows specific amplification of Salmonella DNA, so the highest fluorescence variation in Salmonella DNA was shown. However, E. O157 and negative (NTC) showed only no DNA amplifications. The slight variations are attributed to only non-specific amplification above 35cycles in both E. O157 and negative (NTC). Therefore, the intensity of both E. O157 and negative (NTC) should be equivalent.

  1. To qualify as a point of care testing method; you must demonstrate recognition of unknown pathogen/diagnosis of infection using this device.
  • As you know, the E. O157 and Salmonella are representative food poisioning bacteria. We already tested the specific amplification of Salmonella and non-specific amplification of E. O157.
  • The designed system accommodates two tubes for the proof-of-concept study, but additional extra tube array can be expanded to analyze many species of bacteria.

  1. Or title should be modified after removing the words "point of care diagnosis".  

- Point-of-care diagnosis is meaning of field tests without transfer of target samples to a genomic laboratory. Sometimes, point-of-care diagnosis means testing in small local clinic. Therefore, the developed device is potential to be applied in POC diagnosis due to the automated fast PCR system including sample preparation.
